# Estimation of Elbow Wall Thinning Using Ensemble-Averaged Mel-Spectrogram with ResNet-like Architecture

**DOI:** 10.3390/s22113976

**Published:** 2022-05-24

**Authors:** Jonghwan Kim, Byunyoung Chung, Junhong Park, Youngchul Choi

**Affiliations:** 1School of Mechanical Engineering, Hanyang University, 222 Wangsimni-ro, Seongdong-gu, Seoul 04763, Korea; givemeletter@hanmail.net (J.K.); parkj@hanyang.ac.kr (J.P.); 2Smart Structural Safety and Prognosis Research Division, Korea Atomic Energy Research Institute, 111 Daedeok-daero 989 Beon-gil, Yuseong-gu, Daejeon 34057, Korea; cby@kaeri.re.kr

**Keywords:** wall thinning, loop test, convolutional neural network, vibration characteristics, ensemble average, residual block, mel-spectrogram

## Abstract

An elbow wall thinning diagnosis method by highlighting the stationary characteristics of the operating loop is proposed. The accelerations of curved pipe surfaces were measured in a closed test loop operating at a constant pump rpm, combined with curved pipe specimens with artificial wall thinning. The vibration characteristics of wall-thinned elbows were extracted by using a mel-spectrogram in which modal characteristic variation shifting can be expressed. To reduce the deviation of the model’s prediction values, the ensemble mean value of the mel-spectrogram was used to emphasize stationary signals and reduce noise signals. A convolutional neural network (CNN) regression model with residual blocks was proposed and showed improved performance compared to the models without the residual block. The proposed regression model predicted the thinning thickness of the elbow excluded in training dataset.

## 1. Introduction

Pipe wall thinning due to high-temperature and high-pressure flow is a kind of failure in nuclear power plants, and several serious incidents due to wall thinning were reported in 1986 and 2004 [1,2,3]. Since the pipe wall thinning phenomenon by flow accelerated corrosion (FAC) poses a threat to safety, major countries have developed codes to predict pipe wall thinning [4,5,6]. Reliable analysis and decisions for the inspection are made according to the codes to prevent pipe rupture. Piping inspection and replacement management of pipes selected by decision are carried out by periodic inspection. The ultrasonic test (UT) is mainly used to measure the pipe wall thickness in periodic inspection [7,8]. Since thickness measurement errors with UT can be caused by the operators and the inspection environment, how to reduce the measurement error of UT [9,10] and other types of non-destructive testing (NDT) inspection methods [11,12] have been studied. Since the traditional NDT inspection characteristics are point-by-point measurements, it is necessary to measure all points at regular intervals in a specific pipe area, which takes significant time. Periodic inspections have a limited number of pipes that can be inspected within the inspection. For piping excluded from the periodic inspection list, the inspection cycle of the pipe may be prolonged, and there is the possibility of missing some failures between the successive inspection cycles [13].

Condition-based maintenance (CBM) has been developed to make up for the weakness of periodic inspection in preventing unexpected accidents. CBM in nuclear power plants can serve to provide better prognostics and reduce excessive maintenance by condition monitoring [14]. Studies to monitor the wall thinning state with attached sensors have been proposed. A method with an ultrasonic waveguide system [15] is difficult to use in an area in which it is hard to install a wave guide. Safety problems may occur due to the high current in the direct-current potential drop (DCPD) method [16]. We intend to propose a method to monitor the state of wall thinning using an accelerometer that is relatively easy to install. Loop tests were prepared by fabricating wall thinning specimens with different amounts of thinning, following the actual shape of the elbow replaced after use in a nuclear power plant. Using a loop test, combined with the prepared specimen, the vibration characteristics were examined according to thinning through the acceleration signals measured under constant operating conditions. As in [17], it was observed that the higher the natural mode order, the larger the change in the natural frequency according to thinning amount.

This study proposes a method of predicting elbow wall thinning without selecting the frequency components of the accelerometer signal related to the wall thinning. The CNN model, which utilizes a convolutional filter to recognize the image features, has remarkable performance in terms of image recognition [18,19,20,21]. Vibration signals were imaged and used as the input data to realize the advantages of the convolutional neural network (CNN) model. Therefore, one-dimensional accelerometer signals measured at the elbow need to be translated into two-dimensional image input data. By applying a mel-filter to the vibration signal, this was converted into a mel-spectrogram in the form of an image. The mel-spectrogram consists of a frequency axis and a time axis. The characteristics of the spectrum, which have a larger change in the high-frequency region than in the low-frequency region according to the amount of wall-thinning, were factorized into fewer components using a mel-filter as the frequency axis of the mel-spectrogram. The vibration changes over time during loop operation are expressed through the time axis of the mel-spectrogram. The mel-spectrogram converted from the vibration signal was input and used by a CNN model to learn the changes in vibration characteristics due to wall thinning. The signal during the loop operation is a stationary signal and a kind of random signal containing noise [22]. Since it is difficult to observe the main characteristic components when the noise signal is large, the noise signal of the mel-spectrogram data is reduced using the ensemble average, making it easier for the CNN model to learn the vibration characteristics. Conventional convolution filters have a structure that directly performs convolution operations on data transferred from the previous layer. In the residual blocks structure seen in [23], the data transferred from the previous layer are separately stored through skip connection, and summed with the data to which the convolution filter is applied. When the convolution filter learns about the residuals with the original data using the residual block, the model optimization is easier and the gradient vanishing phenomenon can be reduced. The proposed model was designed by applying the residual block to the existing Lenet-5 model [24]. A deep learning model can be divided into a classification model and a regression model according to the output layer [25,26], but the classification model has a limitation, as it outputs only one of the labels used for model training. A thinning diagnosis method using an accelerometer was proposed, and the thinning state was diagnosed through a classification model [27]. The method classifies the data into three kinds of thinning state, but there is a limitation in that other thinning states cannot be predicted. Unlike the classification model, the regression model has high scalability because the output value can be expressed as a real number, allowing for output in addition to the value used in training. A methodology using regression model was studied to predict the wall thinning rate using various factors [28], but uncertainty may exist in the wall thinning prediction of the final state due to the accumulation of wall thinning rate errors predicted at each state. In addition, the nonlinear behavior of FAC is expressed based on the measured thickness and various environmental factors through the codes, but it is difficult to determine the actual cause of pipe wall thinning. This study proposed a regression model for predicting wall thinning through a vibration signal for more effective management by monitoring elbow wall thinning.

## 2. Materials and Methods

### 2.1. Fabrication and Experiments of Pipe Wall Thinning

The stress is induced due to the high-temperature and high-pressure flow in the pipes, resulting in a thinning that mainly occurs in the curved pipe. The cross-section of the 90-degree curved pipe, which was replaced after use in the nuclear power plant, was measure, and it was confirmed that the maximum reduction occurred 34 degrees from the flow outlet. A thinning curved pipe was designed, as shown in Figure 1a, by simulating the actual shape of the thinning curved pipe. In order to measure the vibration characteristics according to the thinning amount, six specimens with the maximum thinning in the curved pipe—0% (non-thinning), 14%, 28%, 42%, 56%, and 70%—were produced. Since the thinning specimens were manufactured through machining, the actual thinning thicknesses are different from the designed thickness, as shown in Table 1. The minimum thicknesses of the actual specimens were set to the label values of each data. To combine the specimens with the loop, a straight pipe with a nominal size of 8-inch Sch20 (external diameter 219.1 mm, thickness 6.35 mm) was used to manufacture thin-walled curved pipes. A234 WPB (A106B, carbon steel) was used as the material.

The wall-thinned specimens were combined with the loop, as shown in Figure 1b, and the experiments were conducted under constant pump operation. The pump was operated at 800 rpm and the demineralized water was transported in the loop. The water was pressurized at 1 bar and the flow rate was 128$\;m^{3}/h\;$. As shown in Figure 2, three accelerometers were attached on the outer surface of the curved pipe to obtain the data. The accelerometers measured the data for 600 s using B&K’s type 4375 product, and the sampling frequency of the accelerometers was set to 16,384 Hz to measure the data for each specimen.

The measured acceleration signal was converted into a mel-spectrogram to extract the features due to wall thinning. Figure 3 shows the mel-spectrograms transformed from 4 s acceleration signals measured by an accelerometer located 22.5 degrees away from the flow outlet. A mel-filter was used for the transformation, with 256 components. The data were measured for each specimen under the condition that the pump was operating at a constant rpm. In the frequency range of 600 Hz or less, the pump excitation characteristics were observed. In the frequency range of 600 Hz or more, flow-induced excitation occurred and the mel-spectrogram changed according to the wall thinning. The CNN model was applied to learn the change in the mel-spectrogram according to the wall thinning.

### 2.2. Data Preparation for Predicting Elbow Wall Thinning

To predict the thinning thickness by applying a deep learning model, the process of selecting input data, extracting data features, designing the model, and selecting output data is necessary, as shown in Figure 4. In the data feature extraction step, it is necessary to analyze the data to understand the vibration characteristics and set the main frequency range to be used as the feature. To simplify this process, a method for predicting the thinning thickness through the mel-spectrograms using the characteristics of the frequency domain and the time domain was studied.

The input data comprise an accelerometer signal measured through a loop test, and the feature extraction process for the input data is simplified by mel-spectrogram transformation. The measured data have a sampling frequency of 16,384 Hz, so when the frequency resolution is set to 1, the maximum frequency is 8192 Hz. Since each piece of data has a size of 8192 in the frequency domain, the size of the input data becomes too large and unnecessary features can affect model learning. A mel-filter was applied to increase the frequency resolution and reduce the data size while preserving the wall thinning characteristics of the frequency component. If the spectrum has a range of 600 Hz or higher, the resonance frequencies are shifted according to the thinning thickness and the level of the spectrum changes over a wide frequency range. The mel-filter has a wider bandwidth with a higher frequency range. The mel-spectrogram with the mel-filter applied filters the wider frequency range characteristics into a single component in the higher frequency range.

Data of 4 s in length were used to recognize the change over time. When the noise signal is too large, it is hard for the CNN model to learn the data features. The ensemble average value of 10 consecutive pieces of data was used to increase the learning performance of the model by reducing the noise signal. Since CNN is generally used for square-shaped image data, 256 (frequency components) × 257 (time components)-sized mel-spectrograms (where a mel-filter with 256 components was applied to 4 s acceleration signals) close to a square shape suitable for CNN model learning were used as input for the model. The output data were set as the thinning thickness.

### 2.3. CNN Model Development for Predicting the Wall Thinning Thickness

A CNN model was developed through input and output data selection and model architecture design. The CNN model to which the structure of the residual block was applied was proposed by adding skip connection to the Lenet-5 model, as shown in Figure 5, to improve learning about the stationary signal during operation. The model is composed of an input layer, a first residual block, a first average pooling layer, a second residual block, a second average pooling layer, a flatten layer, a first fully connected (F.C) layer, a second F.C. layer, and an output layer, in that order.

The input layer, which is the first layer of the model, receives the mel-spectrograms prepared in the previous section as input data. In the range of 600 Hz or less, the mel-spectrum value is larger than the value of range of 600 Hz due to pump excitation. When training the deep learning model, the model is dominated by the range of relatively large values. This means that the model struggles to learn the features of a range composed of small values. The logarithm of the mel-spectrogram was calculated to scale the small value as relatively large and the large value as relatively small. The model using the logarithm of the mel spectrogram was not dominated by values in the range of 600 Hz or less. As shown in Equation (1), for each piece of data, the log was normalized based on the maximum and minimum values of the total data, and then used as the input data.
(1)$$x^{\prime }=\frac{\left[\log_{2}x-\min\left(\log_{2}x_{total}\right)\right]}{\left[\max\left(\log_{2}x_{total}\right)-\min\left(\log_{2}x_{total}\right)\right]}$$

x denotes original data and $x^{\prime }$ denotes the normalized data. The normalized input data are transferred to a residual block consisting of a layer that transmits the input data through skip connection and a layer that calculates the convolutional filter on the input data. After calculating the two layers of the residual block in parallel, the two calculated results are added and the activation function is applied. In the layer that calculates the convolutional filter, the weights of the filter bank are multiplied and the biases are added, as shown in Equation (2).
(2)$$y_{k}=\overset{m}{\underset{c=1}{\sum }}w_{c,\;k}*x_{c}+b_{k}\;k=1,2,\ldots n$$

$x_{c}$ denotes the *c*-th channel of the input data vector; $y_{k}$ denotes the *k*-th channel of the output data vector. *m* is the total channel number of the input data. *n* is the total channel number of the output data. $w_{c,\;k}$ is the weight vector of the *k*-th filter channel concerning the *c*-th channel of the input data vector and $b_{k}$ is the *k*-th filter channel of the bias vector. * denotes the convolution operator of the vectors. Through the convolutional filter calculation, the data of *m* channels are transformed into the data of *n* channels.

The value that has passed through the convolutional filter goes through an activation function. As the activation function, the ReLU function in Equation (3) was used to add non-linearity [29].
(3)$$\sigma \left(y_{k}\right)=\max\left(0,y_{k}\right)$$

*σ* denotes the activation function ReLU. In order to add the data that were delivered through the skip connection and the data that were calculated with the convolutional layer, the two types of data must have the same size. If the number of channels in the convolution filter is different from the number of channels in the input data, ${w^{\prime }}_{c,k}$ is calculated so that the input data with *m* channels are stored and output as data with *n* channels. Then, the results are added to Equation (3) with *n* channels and an activation function is applied, as shown in Equation (4).
(4)$$z_{k}=\sigma \left(\sum_{c=1}^{m}{w^{\prime }}_{c,\;k}*x_{c}+\sigma \left(y_{k}\right)\right)$$

$z_{k}$ denotes the output value of a residual block. Values that have gone through the residual block and the activation function are transferred to an average pooling layer. An average pooling layer extracts the average at specific intervals to reduce the size of the data, preventing overfitting by causing data loss [30].

Data that have passed through the average pooling layer are arranged in a one-dimensional form through the flatten layer and then connected to the FC layer. The FC layer calculates the weights and biases of the previous layer’s data, and transfers them to the next layer, as shown in Equation (5).
(5)$$y_{j}=\overset{m}{\underset{i=1}{\sum }}w_{ij}x_{i}+b_{j}$$

*m* denotes the total nodes of input data. $x_{i}$ denotes the *i*-th node of input data and $y_{j}$ denotes the *j*-th node of the output data. $w_{ij}$ denotes the weight connecting the $x_{i}$ to the $y_{j}$ and $b_{j}$ denotes the bias scalar of the $y_{j}$. After Equation (5), the ReLU activation function was applied to add nonlinearity.

After the fully connected layer, one node on which the value of the thinning thickness was output was disposed of in the last output layer. To design a regression model, a linear function is applied to the activation function to output the real number and a mean square error (MSE) is applied to the loss function of the model. The Adam function is applied to the optimizer to update the parameters and the learning rate of the optimizer is set to 0.0003.

The detailed architecture of the proposed model is shown in Table 2. The input data of 256 × 257 size are received and delivered to the first residual block. The skip connection layer that delivers the data as they are while increasing the data’s channel is added to the data that were calculated with the 5 × 5 size 6-channel convolution filter in the residual block. The ReLU activation function and the first average pooling layer are then applied. One average value is extracted for each region of 2 × 2 in the average pooling layer and transferred to the second residual block. The 5 × 5-sized 16-channel convolution filter is used in the second residual block. The second average pooling layer extracts one average value for each 2 × 2, as in the first average pooling layer. Three-dimensional data are converted to one-dimensional data through the flatten layer. The flatten values are linked to the fully connected layers. The number of nodes is set to 120 in the first fully connected layer and the number of nodes is set to 80 in the second fully connected layer. Finally, the thinning thickness value is output from the output layer, which is composed of one node.

## 3. Results

### 3.1. CNN Model Evaluation Using Experiments Data

#### 3.1.1. Loss of CNN Model

For the regression model, since a performance evaluation cannot be performed with accuracy when determining whether the prediction label is correct, the model was evaluated through the loss calculated from the difference between the prediction value of the model and the target value. A smaller loss means that the prediction value is closer to the target value. Passing the entire input dataset forward and backward through the model once is called an epoch. The parameters are updated to reduce the loss value at each epoch. However, excessive epochs may over-fit the model with training data, resulting in a larger loss value for the data not used in training. The loss for validation dataset was used so that the model could be generalized to prevent overfitting. A total of 20% of the data for each specimen were used as validation data, and the model parameter of the epoch that had the smallest loss for the validation dataset during 50 epochs training, was stored. The predictive performance of the stored model on the test dataset was evaluated.

#### 3.1.2. Model Evaluation Using Test Dataset

The prediction performance of the unseen data was evaluated through the model trained with mel-spectrogram characteristics according to the elbow wall thinning. Of the data for each specimen, 60% were used as training data. Half the remaining data were used as validation data and half were used as test data. Figure 6 shows the results of evaluating the test data through the model trained by the ensemble-averaged mel-spectrograms and non-averaged mel-spectrogram. The average of the prediction values is expressed as the height of the bar and the standard deviation of the prediction values is expressed as the tip at the end of the bar. When the deep learning model was trained with mel-spectrograms converted from acceleration signals, it was possible to estimate the thinning thickness of unseen data by training the characteristics according to the thinning thickness change. In comparison with using the non-averaged data, the prediction results of the model using the ensemble average values were confirmed to be closer to the actual specimen thickness and the standard deviation is smaller than the prediction results for the model using original data.

The results of the model using a deep neural network (DNN), CNN without residual blocks, and the proposed CNN were compared, as shown in Table 3. In the case of the proposed model, the MSE loss value for the test dataset was the best.

Additionally, the proposed model to verify the prediction performance for missing data was trained with mel-spectrograms with one specimen data missing from all specimens. Models trained with five specimen data evaluated the test data with six specimen data. Table 4 shows the prediction results for the missing specimen data by the trained model. The prediction values for missing specimens are close to the actual thickness. When ensemble-averaged data were used, the predicted values were more similar to the actual values.

## 4. Conclusions

To measure characteristic changes in an acceleration signal during loop operation according to the wall thinning of the curved pipes, loop tests were performed, combined with six artificially curved pipe specimens imitating the actual wall thinning. In the spectrum transformed from the measured acceleration signal, the excitation of the pump was observed in the range of 600 Hz or less and the flow-induced excitation was observed in the region of 600 Hz or more. The spectrum level changes according to the wall thinning thickness and the shifts in the resonance frequencies occurred in the range of 600 Hz or more. The acceleration signals were transformed into mel-spectrograms using a mel-filter, so that the changes in spectrum levels over a wide band of high frequency and in resonant frequency could be expressed with small-frequency components. The signal during loop operation is a random signal in a stationary state. The ensemble mean value was used to reduce the deviations in the noise signal and used for model training. A CNN regression model with residual blocks was proposed to learn the characteristics of stationary signal by using the mel-spectrograms, and the proposed model has improved loss values compared to DNN and CNN. It was possible to learn the vibration characteristics according to the thinning thickness using the features of the entire frequency domain by omitting the feature extraction process that extracted the values of a specific frequency domain. To verify the prediction performance for the missing data, a training dataset with one specimen datum missing from all specimen data was used to train the model. The trained model output the thinning thickness of missing specimen close to the actual value. Through the regression model, it was possible to predict the elbow thinning thickness using one accelerometer signal, and it is expected that it can be used as a thinning monitoring method.

## Figures and Tables

**Figure 1 sensors-22-03976-f001:**
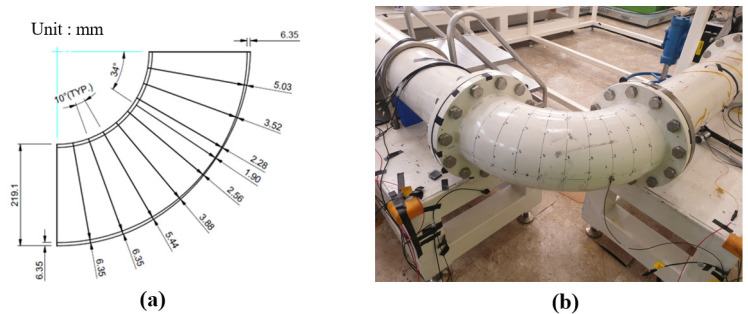
(**a**) Section view of a wall thinning curved pipe with 30% thickness at a 34-degree position and (**b**) loop combined with a thinning elbow.

**Figure 2 sensors-22-03976-f002:**
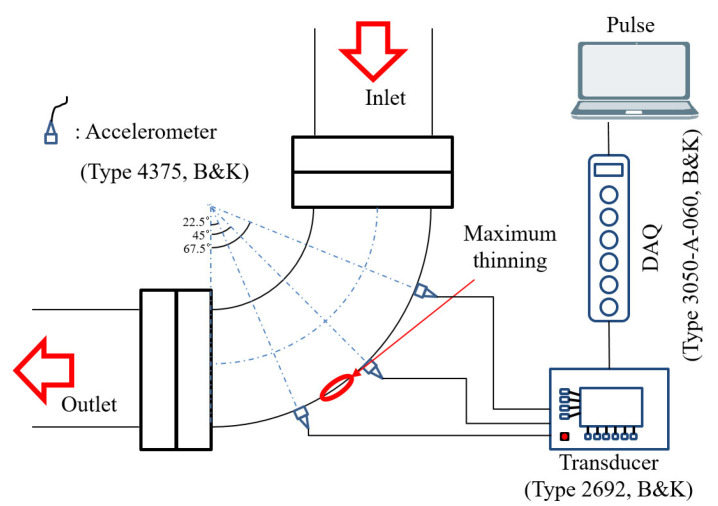
Vibration signal measurement system and accelerometer positions at the surface of the curved pipe.

**Figure 3 sensors-22-03976-f003:**
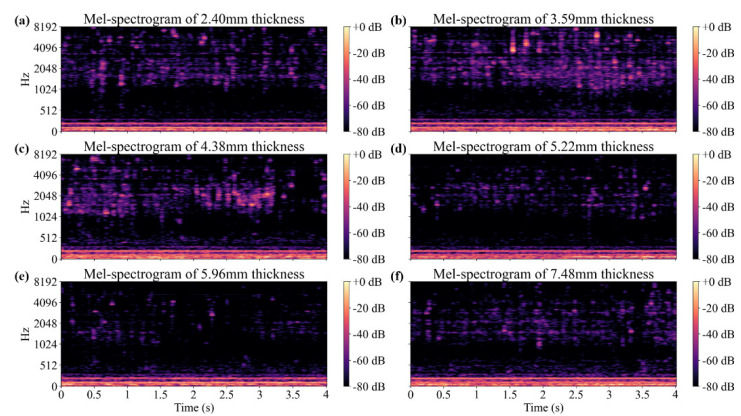
Mel-spectrograms calculated with 256 mel-filters from the acceleration measured at 22.5 degrees of (**a**) 2.40 mm, (**b**) 3.59 mm, (**c**) 4.38 mm, (**d**) 5.22 mm, (**e**) 5.96 mm, and (**f**) 7.48 mm thickness specimens.

**Figure 4 sensors-22-03976-f004:**
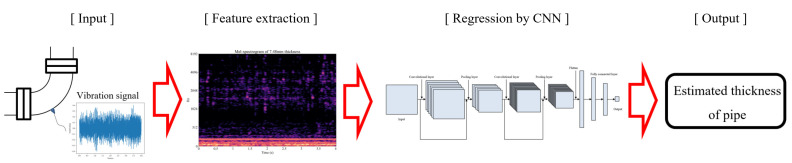
Wall thinning monitoring process using a deep learning model, from signal measurement to thickness estimation.

**Figure 5 sensors-22-03976-f005:**
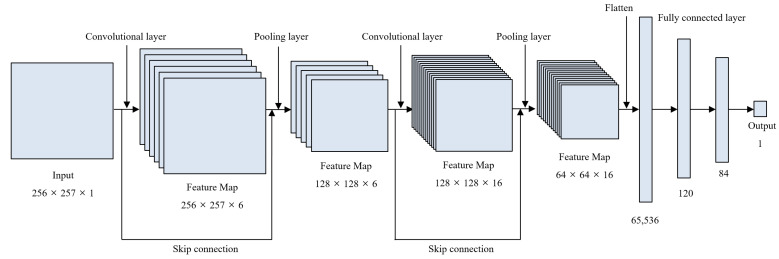
Proposed CNN model architecture using residual block.

**Figure 6 sensors-22-03976-f006:**
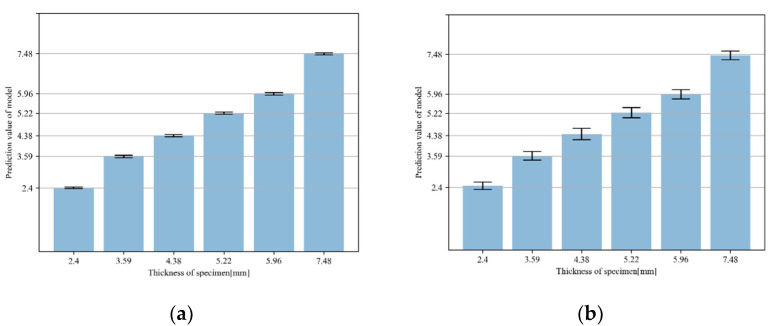
Prediction values of test dataset through model trained by the (**a**) proposed method and (**b**) traditional method, with the accelerometer located at 22.5 degrees.

**Table 1 sensors-22-03976-t001:** The thinning thickness of six fabricated specimens.

Max. Thinning Ratio	Min. Thickness of the Elbow
0%	7.48 mm
20.3%	5.96 mm
30.2%	5.22 mm
41.4%	4.38 mm
53.0%	3.59 mm
77.9%	2.40 mm

**Table 2 sensors-22-03976-t002:** Detailed architecture of proposed model.

Layer	Input Shape	Output Shape
Input layer	(256, 257, 1)	(256, 257, 1)
Skip connection 1	(256, 257, 1)	(256, 257, 6)
Convolutional layer 1 (ReLU)	(256, 257, 1)	(256, 257, 6)
Add 1 [Conv1, Skip1] (ReLU)	(256, 257, 6)	(256, 257, 6)
Average pooling layer 1	(256, 257, 6)	(128, 128, 6)
Skip connection 2	(128, 128, 6)	(128, 128, 16)
Convolutional layer 2 (ReLU)	(128, 128, 6)	(128, 128, 16)
Add 2 [Conv1, Skip1] (ReLU)	(128, 128, 16)	(128, 128, 16)
Average pooling layer 2	(128, 128, 16)	(64, 64, 16)
Flatten layer	(64, 64, 16)	(65, 536)
FC layer 1 (ReLU)	(65, 536)	(120)
FC layer 2 (ReLU)	(120)	(84)
Output layer (Linear)	(84)	(1)

**Table 3 sensors-22-03976-t003:** MSE loss comparison for each dataset according to the deep learning model.

**Model**	**Mean Squared Error**		
	**Training Data**	**Validation Data**	**Test Data**
DNN	0.003665	0.005296	0.005493
CNN	0.000440	0.001770	0.002042
CNN + residual block	0.000238	0.001186	0.001244

**Table 4 sensors-22-03976-t004:** Mean and standard deviation of the prediction values for the missing specimen of the model trained with the raw data and the ensemble-averaged data without one specimen.

Missing Specimen in Training Dataset	Non-Averaged Data		Ensemble-Averaged Data	
	Mean	Standard Deviation	Mean	Standard Deviation
3.59	3.34	0.236	3.49	0.205
4.38	4.75	0.279	4.38	0.215
5.22	5.13	0.279	5.26	0.200
5.96	5.75	0.223	5.94	0.178

## Data Availability

Not applicable.

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
