# Peer review of "Estimation of Elbow Wall Thinning Using Ensemble-Averaged Mel-Spectrogram with ResNet-like Architecture"

_sensors, 2022, doi:10.3390/s22113976_

Round 1
Reviewer 1 Report
Comments and Suggestions for Authors
- The reviewer would kindly ask authors to add some scientific added value or scientific findings of this work during the revision process. (Abstract, Introduction, Example, or Conclusions)
- It seems that only one practical case study of a wall thinning curved pipe is provided to confirm the validity of the proposed method. The author should demonstrate the effectiveness of the proposed method from the other (experimental or industrial) thin-walled curved pipe case(s) in this paper.
- The reviewer would kindly the authors to add more references (and try to avoid self-citing) from the last 5 years, and the grammar of some sentences in the manuscript needs to be confirmed/proofread during the revision process.
- All the symbols (of operators) used in the manuscript should be included in the nomenclature.
- The authors should clarify why, what, and how to use mel-spectrogram combined with CNN in this manuscript to reduce the deviation of the prediction values of the model in this manuscript. Also, how do the authors make sure that the random signal in a stationary state to meet and learn the stationary property of CNN (or DNN) in conjunction with mel-spectrogram?
Reviewer 2 Report
This paper presents a data-driven method to estimate the wear of the walls on a pipe using the mel-spectrogram and a convolutional network. In general, the contribution is not clear. However, I find the paper interesting, and it may have some merit, but it has to be considerably improved.
* In my view the paper is not well-motivated. The motivation has to be further strengthened, why the thinning of the pipe walls are important to be monitored? There is a mention of a safety-critical system, but in practice, how long does it take to reach a significant thinning of a pipe? This is not sudden, the maintenance/replacements can be programmed.
* The title and abstract are about a pipe. However, the problem addressed, the experiments, and all the results are for a pipe accessory, an elbow. I suggest being more specific.
* The literature review has to be improved, most of the literature is outdated. Other Neural Networks have been applied for similar tasks recently, e.g. Leak diagnosis in pipelines using a combined artificial neural network approach, Control Eng.Pract.; An intelligent model to predict the life condition of crude oil pipelines using artificial neural networks, Neural Computing and Applications; among others. What would be the difference?
* Include an analysis of recent 2019-22 literature to justify the contribution.
* What fluid is being transported in the pipe for the experiments? Does a different fluid affect the results? Could the method be adapted for a different fluid? What are the conditions of fluid? Please discuss.
* The method itself does not seem to be practical to be used in a real environment. You needed three sensors for a simple elbow plus the training. How many sensors would you need for a real pipe network?
* One of the main problems in NN is the training. In this case, only six 90 degree elbows with six thickness variations were used for training. 60% for training, 20% for validation and 20% for tests. For completeness, the algorithm must be tested with new data obtained for an elbow with a thickness not originally used for training.
* Other scenarios should be tested as well. What if the fluid condition changes? What about the pressure, how does it influence the results of the method?
* A comparison with a relevant up-to-date algorithm would be acknowledged.
* The language must be improved. There are typos and other grammatical errors along the document.
Round 2
Reviewer 1 Report
I am satisifed with the replies from the authors of this paper.
Author Response
Thanks you for your acceptance.
Reviewer 2 Report
Some of my comments were where not addressed in the revision.
My first comment was about motivation. Why do the pipes have to be monitored in the way the authors propose? Even in the case of a safety-critical system, there are programmed periodic maintenance and the walls of the pipe (or pipe elbows) do not degrade from one day to another. The authors' response is "some are operated longer than the design operating years", "periodic inspection (about 18 months)", this is false for safety-critical systems such as a nuclear plants. The inspection and maintenance are taken very seriously. However, apart from what is written in the response letter, no action was taken by the authors in the paper. The motivation has not been improved in the manuscript.
There is no mention in the manuscript of the fluid transported in the pipe. There is no discussion about changing the fluid and how it affects your method. The is no discussion about changing other parameters such as the pressure, and pipe material, among others.
There is no discussion about the number of sensors required in a real situation or industrial environment. In fact, the overall practicability of the solution proposed is questionable.
There is no comparison with a recent method.
Round 3
Reviewer 2 Report
I have no more comments